# The Construction of a Footbridge Prototype with Biological Self-Healing Concrete: A Field Study in a Humid Continental Climate Region

**DOI:** 10.3390/ma15238585

**Published:** 2022-12-01

**Authors:** Ronaldas Jakubovskis, Renata Boris

**Affiliations:** 1Laboratory of Innovative Building Structures, Institute of Building and Bridge Structures, Vilnius Gediminas Technical University, Sauletekio 11, 10223 Vilnius, Lithuania; 2Department of Reinforced Concrete Structures and Geotechnics, Vilnius Gediminas Technical University, Sauletekio 11, 10223 Vilnius, Lithuania; 3Laboratory of Composite Materials, Institute of Building Materials, Vilnius Gediminas Technical University, Sauletekio 11, 10223 Vilnius, Lithuania

**Keywords:** biological self-healing concrete, footbridge prototype, field studies, large-scale demonstration, site trials

## Abstract

Biological self-healing concrete (BSHC) offers a sustainable and economical way of increasing the lifespan of structures vulnerable to cracking. In recent decades, an enormous research effort has been dedicated to developing and optimizing the bacterial healing process. Nevertheless, most studies have been carried out under laboratory conditions. To verify the effectiveness and longevity of the embedded healing systems under normal service conditions, field studies on BSHC structures must be performed. In the present study, BSHC beams were designed as a structural part of a prototype footbridge. To select the optimal BSHC mix composition, a series of laboratory tests were also carried out. Laboratory tests have shown that the healing ratio in BSHC elements under rain-simulating healing conditions was several times higher in comparison to control specimens. Based on the laboratory results, the BSHC mix composition was selected and applied for structural bridge beams. To the best of the authors’ knowledge, the present study reports the first application of BSHC in a prototype footbridge. The long-term data gathered on the healing process in a humid continental climate zone will allow the benefits of biological self-healing to be quantitatively evaluated and will pave the way for the further optimization of this material.

## 1. Introduction

Concrete is a versatile structural material best known for its high compressive strength, excellent formability and low cost. Recent developments in concrete technology have further broadened the application of this outstanding material. For example, concrete may feature ultra-high bending strength and toughness, the ability to self-consolidate, extreme lightness or the ability to heal cracks itself. Its primary deficiency, however, remains its low tensile strength, making it susceptible to cracking. Once a crack opens, water, oxygen and salts easily penetrate inside, causing corrosion of reinforcement, deterioration of concrete and, consequently, shortening the service life of the structure. Cracking is the most detrimental defect in concrete structures; bridges and underground and water-retaining infrastructures are among the most vulnerable objects [1].

One of the most promising technologies to inhibit the detrimental effect of cracks is the development of self-healing concrete. The idea of self-healing involves the recovery of material integrity without any human intervention, thereby saving on maintenance and repair costs. A bacterial approach which is based on the utilization of calcium carbonate-precipitating bacteria has emerged as one of the most sustainable and effective techniques in the development of self-healing concretes [2].

In recent decades, an enormous research effort has been dedicated developing and optimizing the bacterial healing process. The identification of suitable bacteria strains, efficient microbial metabolic processes and bacteria encapsulation techniques, along with the optimization of concrete composition, have been comprehensively studied and are well summarized in several review papers [3,4,5]. However, most studies have been performed on small specimens under laboratory conditions [3]. In addition, the healing process of plain concrete or mortar specimens containing a single crack is commonly studied, while reinforced concrete structures with multiple cracks of different widths, positions and penetration depths are required to consider healing in practice. The self-healing ability of upscaled concrete structures may be inferior in comparison to small laboratory specimens due to the variability of the material parameters and the significant dilution of the healing agent [6]. Another challenge in upscaling laboratory specimens is the high labor costs of preparing the self-healing agent, especially in the production of BSHC [7,8]. Furthermore, environmental conditions may affect the healing process. For example, the temperature and pH of concrete may significantly suppress the viability of bacteria. The internal temperature in massive concrete elements may reach as high as 60 °C [9,10], which can substantially decrease the number of viable bacterial spores [11]. Similarly, bacterial germination and metabolic activity is strongly related to pH levels, which constantly change with concrete carbonation [12]. Moreover, the healing efficiency of BSHC is affected by the concentration of oxygen and both wet-dry and freeze-thaw cycles, all of which depend on the service conditions of the structure [13,14]. Nevertheless, the production and testing of real-scale structural self-healing concrete elements is essential in order to study the effectiveness and longevity of the embedded healing system [3,6,7].

To examine the benefits of bacterial self-healing concrete under real environmental conditions, several field studies have been performed in the Netherlands [15,16], Belgium [7], the United Kingdom [8] and China [10,17]. 

A wastewater treatment tank and water reservoir constructed in the Netherlands [16] had direct contact with water. Both structures comprised BSHC and reference (normal) concrete wall segments. The structures remained uncracked after several years of operation; thus, a quantitative evaluation of self-healing was not possible.

An inspection pit constructed in Belgium [7] featured a BSHC roof slab that was in contact with soil. The slab was constructed 3.9 m below the ground and was accessible for inspection from the underside, where cracks were expected to appear. The roof slab was not tested in a controlled way, and no appearance of cracks was reported. Thus, the healing efficiency was only evaluated from laboratory specimens. 

A large amount of BSHC was applied for a lock channel wall segment in China [17]. In total, 140.4 m^3^ of BSHC was produced and poured in a wall segment with a length of 18 m. Despite the massive application of biological concrete, only several cracks developed between the bottom slab and side wall of the lock chamber. Consequently, the field data gathered on crack healing in a hydraulic environment was rather limited. 

A wall panel with a height of 1.8 m and width of 1 m was constructed from BSHC and tested in open air conditions in the United Kingdom [8]. Contrary to previously discussed BSHC field applications, this panel was tested in a controlled way using a hydraulic jack positioned between the reaction wall and the BSHC wall segment. In total, three cracks appeared at the bottom side of the panel wall. Such a limited number of cracks may be insufficient for rigorous statistical analysis of healing efficiency.

In general, studying the healing efficiency of BSHC in field conditions is quite challenging. If a structure operates under normal service conditions, very few cracks may develop [8,17] or the structure may remain uncracked [7,15,16]. To obtain the maximum amount of data on crack healing efficiency with a limited amount of BSHC, mixed testing methods were applied in this study. BSHC beams were tested in laboratory conditions, controlling both the number and widths of cracks. After laboratory tests, BSHC beams were installed as structural elements in the bridge prototype for the further observation of healing under real environmental conditions. To the best of the authors‘ knowledge, the present study reports the first application of BSHC in a prototype footbridge. A series of laboratory tests simulating the natural environmental conditions allowed for the final composition of structural BSHC beams to be selected. The BSHC beams were designed and produced as structural parts of the prototype footbridge. The long-term data on the healing process gathered under open-air operation conditions will serve as a basis for the further optimization of this material in a humid continental climate region.

## 2. Footbridge Design

The total span of the bridge prototype was 7.2 m (2.8 + 1.6 + 2.8 m), sufficient to pass a small water stream in the Botanical Gardens of Vilnius University (Figure 1). The location of the footbridge inspired the application of biomimetic architectural principles. The shape of the bridge mimics the natural branch growth of *Pinus sylvestris*, the most common tree species in Lithuania. Four tree-like cantilever trusses (two from each side) were designed as the main load-bearing elements of the bridge, while the central part acts as a simply supported beam. This bridge design presents benefits with regard to the installation and replacement of the central bridge part. The bridge having replaceable beams allows us to perform several consecutive studies of healing performance under real environmental conditions, as beams may easily be dismantled and replaced after the study is finished. Moreover, this design enables cracks to be induced in the beams under laboratory conditions before installation on the bridge. 

The bridge prototype was designed for normal pedestrian use. Consequently, the characteristic value of a uniformly distributed load of 5 kN/m^2^ was assumed according to [18]. The most unfavorable situation was obtained by applying this load on the whole bridge surface (7.2 m length and 1.5 m width). Concentrated loads were not taken into account as service vehicles are not carried on the footbridge.

Ultra-high-performance fiber-reinforced concrete (UHPC) was selected as the structural material for the tree-like trusses due to the extremely complex shape of these structures. The strain-hardening behavior of UHPC allowed us to eliminate the use of bar reinforcement, which was critical for the thin and curved web elements. A detailed specification of the material and structural design of the trusses, however, is outside of the scope of the present study. Here, we instead focus on the design and testing of the central BSHC beams. Further sections give more information on the material composition, crack formation and laboratory and field studies of biological concrete elements.

## 3. Production and Testing of Biological Concrete

### 3.1. Laboratory Tests

Laboratory tests on BSHC specimens were conducted in two stages. At the first stage (Series 1), the *B. pseudofirmus* bacteria strain (DSM 8715, the German Collection of Microorganisms and Cell Cultures, Braunschweig, Germany) was used (Figure 2A). At the second stage (Series 2), tests with *B. Cohnii* bacteria (DSM 6307, the German Collection of Microorganisms and Cell Cultures, Braunschweig, Germany) were performed (Figure 2B). These bacteria strains were selected due to their high sporulation yields and their ability to survive in the concrete matrix [19]. The concrete mix listed in Table 1 was used for both test series. This mix design is based on several of our previous studies where healing performance was examined with respect to the chemical composition of cement, bacteria strain, concrete curing temperature and the impact of negative temperatures and protective coatings on bacteria [11,13,20]. A comprehensive study of bacterial viability in concrete indicated that the best bacterial survivability and, consequently, healing ability of BSHC may be obtained by using cements with minimal amounts of zinc and copper oxides [11]. Bacterial survivability may be further increased by using protective coatings from magnesium oxide or styrene-acrylate emulsion [20]. 

Due to its lower concentration of toxic metal oxides, white Portland cement (Aalborg White^®^, Aalborg, Denmark) was used for the production of BSHC. The healing agent was prepared by impregnating expanded clay particles (Liapor GmbH, Hallerndorf, Germany) with 80 g/L calcium lactate pentahydrate solution, 1 g/L yeast extract and 1 × 10^8^ CFU/mL of bacterial (*B. pseudofirmus* or *B. cohnii*) spores, as described in [11]. To protect the bacterial spores, expanded clay particles were additionally coated with styrene acrylate emulsion (Weberfloor 4716, Saint-Gobain, Paris, France) as described in [20]. First, dry components were put into rotating pan mixer (Zyklos ZZ 75 HE, Pemat, Freisbach, Germany) and mixed for 1 min. Water was then added, and the concrete was mixed for two additional minutes.

Reinforced BSHC prisms were used to measure the crack healing process, whereas plain concrete specimens were employed for bacteria viability tests. Here, we report only the results of crack healing in reinforced specimens. As shown in Figure 2A,B, control specimens of the same dimensions were also produced for laboratory tests. For control specimens, the concrete mix as listed in Table 1 was used, but without bacterial spores. After 28 days of water-immersed curing, cracks in reinforced specimens were induced under a three-point loading scheme. Bottom cracks wider than 50 µm were selected for further inspection. Initial crack widths were measured using a stereo microscope (Stemi 305, Zeiss, Oberkochen, Germany) equipped with 5.0-megapixel digital camera (AxioCam ERc 5s, Zeiss, Oberkochen, Germany). For each crack, 8–10 measuring locations were marked, avoiding parallel cracks, missing aggregates and other irregularities. In each location, crack width was measured at three points (Figure 2C), resulting in around 24–30 measurement points in each crack. 

Next, specimens were moved for a healing period of 28 or 98 days to specifically designed rain-simulating basins (Figure 2D). Water was automatically sprayed on the specimens for 30 min two times per day. Water slowly flowed and evaporated from the bottom face of the specimens after each dry cycle. For the rest of the time, specimens were kept in air with an RH varying between 40% and 50%. Such healing conditions were considered to be more realistic in comparison to common healing under water immersion or in 100% RH rooms.

To evaluate crack closure over time, the healing ratio was selected as the main indicator of healing efficiency:*h* = (*w_i_* − *w_t_*)/*w_i_* × 100%(1)
where *w_i_* is the initial crack width and *w_t_* is the crack post-healing width. Here, 100% healing refers to full closure of the initial crack.

Crack widths in the reinforced specimens were repeatedly measured after 28 and 98 days of healing. The obtained variation of healing ratio over time for Series 1 and Series 2 specimens is shown in Figure 3A,B, respectively. Here, continuous lines represent the calculated healing ratio of each separate crack, whereas dotted lines show the average values. An average healing ratio of 14% was obtained for Series 1 specimens (with *B. pseudofirmus* bacteria) after 98 days of regular water spray. Control specimens also exhibited marginal crack closure, with a 3% healing ratio over 98 days. The autogenous healing of plain concrete is a well-known phenomenon and is mostly related to the swelling of concrete, continued hydration of cement and mechanical blocking of cracks by fine particles [3]. It should be noted that the latter component of autogenous healing may have a minor contribution in rain-simulated healing conditions as some cracks in control specimens even exhibited negative healing ratios. Most likely, loose aggregates were washed out from the crack surface, increasing the measured crack width. A similar effect was reported in [17], where biological concrete was in contact with saturated soil. Due to the water pressure, a large amount of white precipitate flowed from the crack. It was concluded that crack healing efficiency may be related to the flow rate of water through the crack.

Similar healing ratios were obtained for Series 2 specimens (with *B. cohnii* bacteria), where healing ratios of 17% and 2% were observed for biological and control specimens, respectively. The healing ratios of Series 1 and Series 2 are compared in Figure 3C. Crack closure in both Series was considerably lower in comparison to our previous study [20], where healing ratios up to 80% were reported. As discussed previously, such differences are mostly related to the different healing conditions: water immersion results in higher healing ratios in comparison to the regular spraying technique that was used in the present study. Nevertheless, surface water spraying was considered to be a more realistic healing condition for an open-air structure such as the bridge prototype discussed here. On the other hand, direct water spraying onto the crack may hinder the microbial-induced calcium carbonate precipitation, underestimating the healing ratio. To obtain representative results, bacterial healing must be examined in field conditions under the normal service of structures. In the next section, the production and testing of the structural biological concrete bridge beams are discussed.

### 3.2. Field Tests

Laboratory tests have shown that comparable healing efficiency was observed using both the *B. pseudofirmus* and *B. cohnii* bacteria strains. Due to its faster sporulation, the *B. pseudofirmus* bacteria was selected for the production of biological concrete bridge beams. Two biological and two control beams were produced for the central span of the bridge. For biological concrete, a similar concrete mix as that listed in Table 1 was used, but with the replacement of 60 kg/m^3^ of cement with silica fume (MICROSILL, Rufax, Kaunas, Lithuania) and the addition of steel fibers (FIBRAG F-WG 35/65, Fibrocev, Sirone, Italy). Silica fume was used as a pigment in the BSHC mix, avoiding the color contrasts between UHPC trusses and BSHC beams, whereas steel fibers were added (78 kg/m^3^) to secure the required shear strength. The concrete mix was prepared using the same rotating pan mixer (Zyklos ZZ 75 HE, Pemat, Freisbach, Germany). The average compressive strength, established in accordance with BS EN 12390 [21], was 26.84 MPa, with a standard deviation of σ = 1.36 MPa. 

The shape and number of reinforcing bars for the beams were selected based on two requirements: (i) sufficient strength to withstand the design loads; and (ii) the formation of 10–15 cracks in the central part of the beam. After several numerical simulations using the ATENA nonlinear finite element software [22], the final height and number of reinforcing bars was selected (Figure 4A). Before installation on the bridge, the beams were tested in the laboratory via a four-point bending test (Figure 4B). Laboratory conditions allowed the load, number of cracks, and crack widths to be controlled during testing. Moreover, the position of cracks and measuring points were precisely marked before installation.

As predicted by the numerical simulation, 12–15 visible cracks formed in the side profiles of the beams (Figure 5A,B). For long-term examination of healing performance, only cracks wider than 50 µm were marked on the bottom surface of the beams and selected for further inspection (Figure 5C). A digital crack microscope was used to measure the initial crack widths. The tested beams were transported to the construction site and installed on the bridge (Figure 6). Crack monitoring began in September 2022. 

Although the healing ratio may not fully characterize the healing process in reinforced concrete elements, such an approach was used due to the position of the inspected beams. Existing methods in water or gas permeability measurement [13,23] may hardly be applied to inspecting bottom cracks. Based on our experience, optical measurement of crack widths is the most reasonable method of evaluating healing efficiency in field conditions.

## 4. Construction of the Bridge

The most challenging process in the construction of the prototype footbridge was the production of the tree-like trusses. The idea of 3D-printing the trusses was excluded due to existing challenges in printing fiber-reinforced concretes. Often, 3D-printed structures suffer from difficult buildability and exhibit a high degree of anisotropy. Moreover, constitutive models for such materials have not yet been developed [24]. Consequently, it was decided to produce four custom-made formworks. Due to the complex and irregular shape of the web elements, four custom-made formworks were produced. The formworks consisted of stiffened plywood sheets and expanded polystyrene EPS80 (100 mm thickness). Polystyrene sheets were cut with a hot wire according to the final shape of the truss and were glued to the plywood with epoxy resin (Figure 7A). After assembly, the formworks were placed on a shaking table. Concrete was poured in two layers and shaken for 60 s each time (Figure 7B). Fresh concrete was covered with plastic film and left for a 14-day curing period. Next, the trusses were transported to the construction site and demolded from the expanded polystyrene (Figure 7C). Good workability and proper compaction of concrete ensured a smooth and aesthetically attractive surface even in the thinner elements of the truss.

The cantilever design of the bridge required a relatively massive foundation. The bridge abutment was placed on four piles with a diameter of 0.4 m and a depth of 1.8 m, sufficient to resist uplifting forces from the cantilever action (Figure 7D). The compressive chord of the truss was joined to the abutment using a stainless-steel base plate, and a tensile chord was connected using the anchorage bars (Figure 7E). The trusses were joined and stiffened by a braced timber frame (45 × 95 mm) that also served as a support for the bridge decking (Figure 7F). Finally, the central biological concrete beams were installed on stainless steel angles and covered with wooden decking boards (Figure 7G). The gaps between the boards ensured that the water would easily flow from the deck, reaching the cracks in the BSHC beams. Such exposure conditions are considered to be close to the normal service conditions of footbridges. Moreover, horizontally placed beams in open air healing conditions will serve as a supplement to existing field studies on BSHC structures [7,8,9,10,15,16,17]. The bridge railings were produced from the same UHPC concrete mix, mimicking the growth of tree branches. The final bridge design perfectly suited the environment of the Botanical Gardens by visually blending with the growth of old trees.

## 5. Concluding Remarks

Biological self-healing concrete offers a sustainable and economical way of increasing the lifespan of structures vulnerable to cracking. This new approach to repair has attracted a tremendous number of research efforts towards developing a suitable and reliable biological healing system. Nevertheless, most of these studies have been carried out under laboratory conditions. 

In the present study, BSHC beams were designed as a structural part of a prototype footbridge. To select the optimal composition of the biological concrete mix, a series of laboratory tests were carried out. The healing process was examined in specifically designed rain-simulating water basins, with dry-wet cycles repeated two times per day. These healing conditions were considered to be more realistic in comparison to the commonly used practices of full water immersion or healing in 100% RH rooms. Laboratory tests have shown that comparable healing ratios were obtained in BSHC specimens using both the *B. pesudofirmus* and *B. cohnii* bacteria strains (14% and 17%, respectively). These healing ratios were several times higher in comparison to control specimens, which exhibited healing ratios of 2–3%. However, the final healing ratios of BSHC elements were significantly lower in comparison to those that have been reported elsewhere from water-immersed specimens [20]. Most likely, lower healing ratios in rain-simulating conditions are related to the wash-out effect of newly formed bacterial precipitation and loose aggregates. Based on the laboratory results, structural BSHC beams were produced for long-term field tests. To maximize the number of cracks that were studied, a mix of testing methods was applied. The BSHC beams were specifically designed and laboratory-tested to obtain 7–10 cracks in the pure bending zone. The cracked beams were then transported to the construction site and installed on the bridge for further observation of healing. Such healing conditions serve as a supplement to existing field studies on BSHC structures. Notably, most field studies have been implemented on BSHC structures in contact with water or soil, whereas open-air healing conditions are examined in the present study.

The upscaling of BSHC specimens is essential for evaluating the healing performance of reinforced concrete elements. Moreover, the healing process may be affected by environmental effects that are very different to ideal laboratory conditions. Freeze-thaw cycling, fluctuations in temperature and humidity, and natural rain or fog may compromise bacterial viability and, consequently, healing efficiency. The prototype footbridge constructed allows the real-time healing process under natural environmental conditions to be studied. The long-term data gathered on the healing process in a humid continental climate zone will allow the benefits of biological self-healing to be quantitatively evaluated and will pave the way for the further optimization of this material.

## Figures and Tables

**Figure 1 materials-15-08585-f001:**
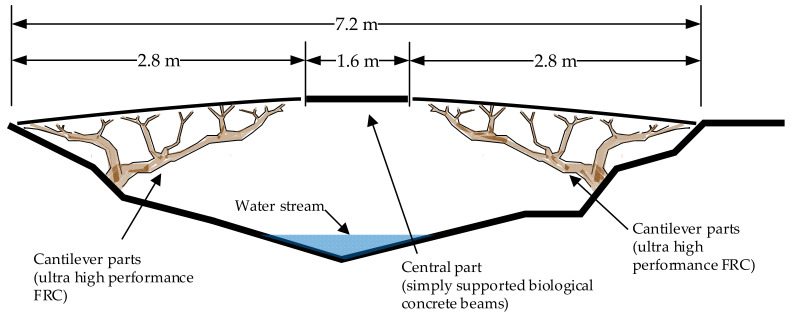
Design of bridge prototype.

**Figure 2 materials-15-08585-f002:**
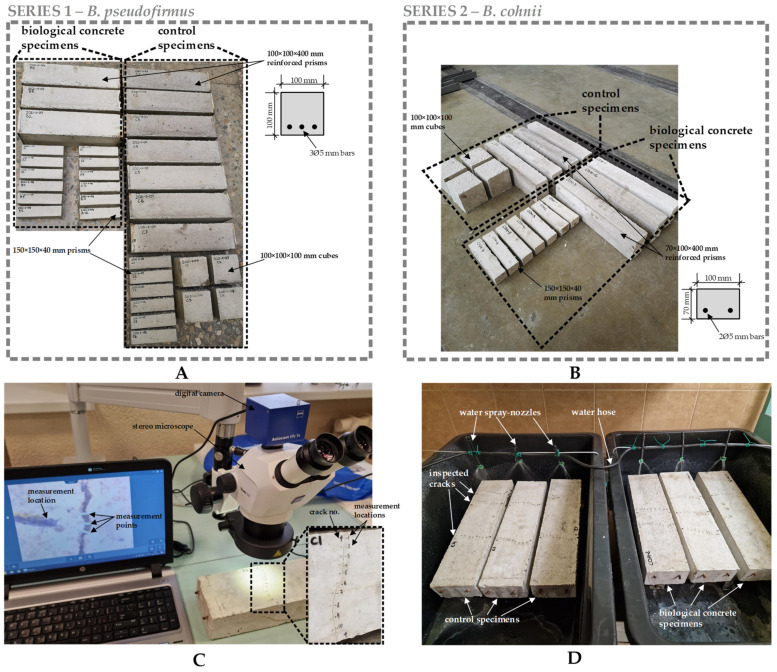
Laboratory tests on BSHC: (**A**) specimens of test Series 1; (**B**) specimens of test Series 2; (**C**) crack measurement scheme; (**D**) healing conditions.

**Figure 3 materials-15-08585-f003:**
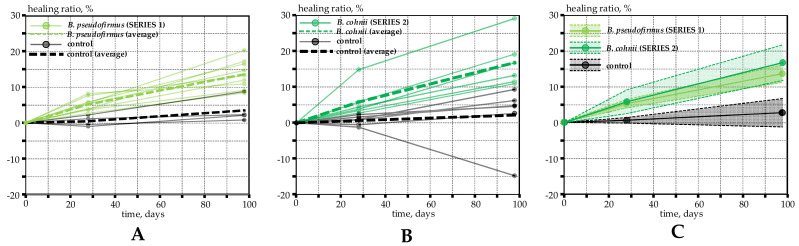
Variation in healing ratio over time: (**A**) results of test Series 1; (**B**) results of test Series 2; (**C**) comparison of healing ratio between test Series 1 and Series 2—continuous lines represent the average values of the calculated healing ratio, whereas shaded areas show 95% confidence intervals.

**Figure 4 materials-15-08585-f004:**
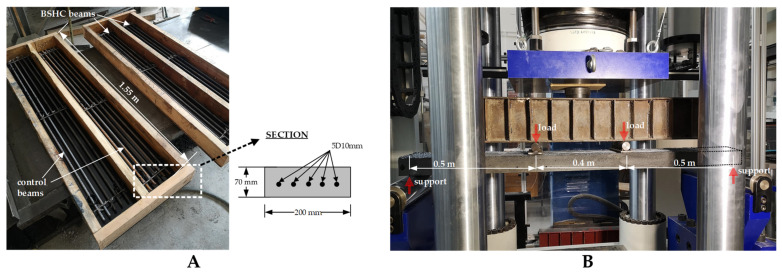
Production and testing of central simply supported beams: (**A**) formwork for biological and control beams; (**B**) load arrangement in four-point bending test.

**Figure 5 materials-15-08585-f005:**
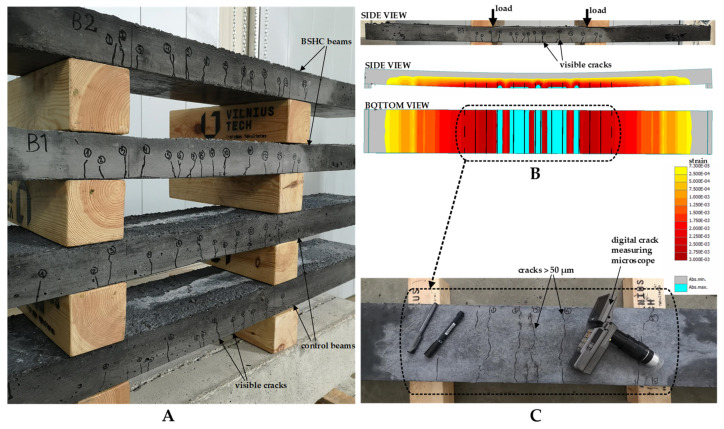
Crack analysis of central beams: (**A**) crack patterns of all tested beams; (**B**) comparison between predicted and experimental crack positions; (**C**) bottom cracks selected for long-term inspection.

**Figure 6 materials-15-08585-f006:**
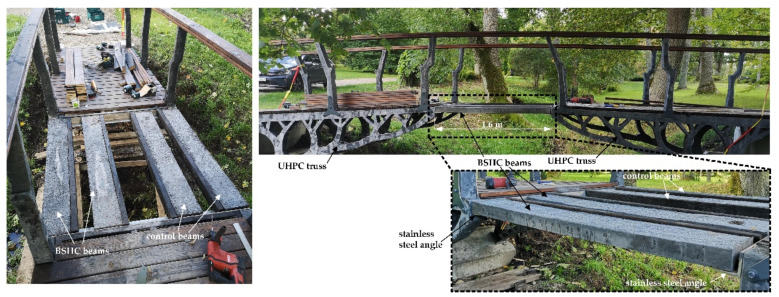
Biological and control beams installed in the bridge for long-term inspection of crack healing.

**Figure 7 materials-15-08585-f007:**
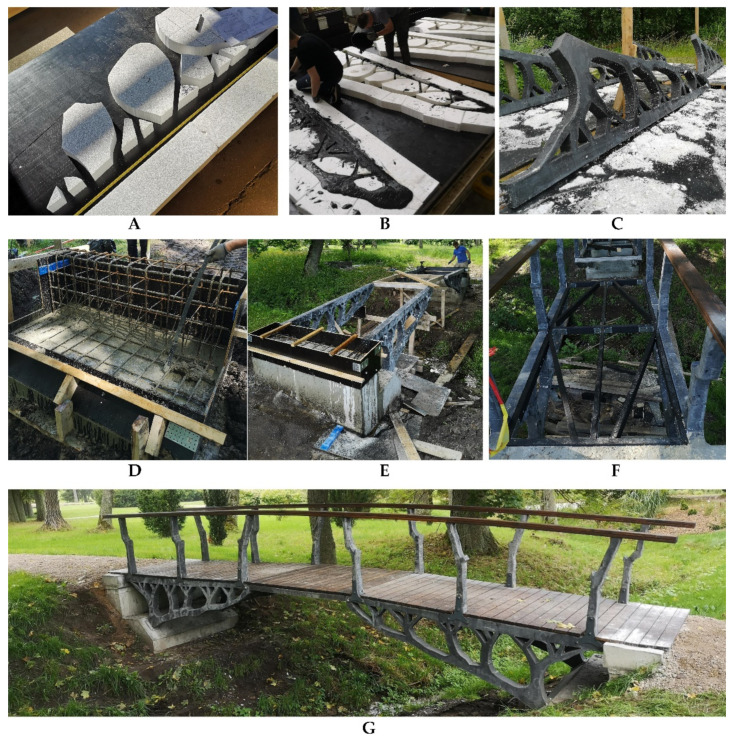
The construction of a prototype footbridge: (**A**) preparation of formwork; (**B**) pouring UHPC mix; (**C**) demolding UHPC trusses; (**D**) construction of foundation; (**E**) installation of UHPC trusses; (**F**) trusses connected with braced timber frame; (**G**) finished prototype bridge.

**Table 1 materials-15-08585-t001:** Composition of BSHC.

Material	kg/m^3^	Mass Percentage
Portland cement CEM I 52.5 R	463	26
Sand (0/4 mm)	855	49
Healing agent—coated expanded clay	270	15
Water	168	10

## Data Availability

The data presented in this study are available on request from the corresponding author.

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
