# Peer review of "The Construction of a Footbridge Prototype with Biological Self-Healing Concrete: A Field Study in a Humid Continental Climate Region"

_materials, 2022, doi:10.3390/ma15238585_

Round 1

Reviewer 1 Report

In general, I think this manuscript makes a good impression and I think it deserves to be published in this Materials - MDPI after the following significant revision:

1. Abstract:

- The objective of this study is not clear.

- The concrete's mixing composition and parameter parameters are unclear.

- Lack of results of this study and discussion information.

2. Introduction:

- Too brief of an introduction. Discuss the concrete mixing composition used in this study in further detail, as well as how the environment affects the performance of the concrete.

- Additional sources relevant to the current study

3. Preparation, fabrication and testing

- Information detail regarding the supplier of the materials used.

- Detail composition (%) of BSHC concrete. Add more information in table 1.

- Clearly describe the mixing procedure, including the tools used, the duration of the mixture, etc.

- Specify any standard method for characterization in this study.

- Figure 4: Missing for Figure 4C. 

4. Construction of the bridge & Concluding remarks:

- Add more references related to this part.

- Discuss the scientific aspect further.

Author Response

In general, I think this manuscript makes a good impression and I think it deserves to be published in this Materials - MDPI after the following significant revision:

We would like to thank the Referee for reviewing our manuscript and for his/her positive evaluation of the study.

  1. Abstract:

-The objective of this study is not clear.

-The concrete's mixing composition and parameter parameters are unclear.

-Lack of results of this study and discussion information.

As suggested, we revised the abstract clarifying the objective and the main outcomes of the study. All changes in the revised version of the manuscript are highlighted in yellow.

  1. Introduction:

-Too brief of an introduction. Discuss the concrete mixing composition used in this study in further detail, as well as how the environment affects the performance of the concrete.

-Additional sources relevant to the current study

We thank the Referee for the important comment. We have improved the introduction section discussing the environmental impact on bacterial self-healing concrete (BSHC). We also discussed the existing field studies on BSHC as well as limitations of these studies. We also added stronger motivation for the current field study in the Introduction section.

  1. Preparation, fabrication and testing

-Information detail regarding the supplier of the materials used.

-Detail composition (%) of BSHC concrete. Add more information in table 1.

-Clearly describe the mixing procedure, including the tools used, the duration of the mixture, etc.

-Specify any standard method for characterization in this study.

- Figure 4: Missing for Figure 4C.

We agree that the manuscript was missing some important details. As suggested, we added the suppliers of the used materials and equipment. We added more information in Table 1 and described the mixing procedure in detail (Section 3.2). We also specified the standard used to determine concrete compressive tests (Section 3.2). Figure 4 was corrected.

  1. Construction of the bridge & Concluding remarks:

-Add more references related to this part.

-Discuss the scientific aspect further.

We thank the Referee for remark. As suggested, we have added some relevant references in the Section 4 “Construction of The Bridge”. We discussed the scientific aspect of these demonstrative project. We also revised the conclusions focusing on the findings from the current study.

Reviewer 2 Report

The paper presents “The construction of a footbridge prototype with biological self-healing concrete: A field study in a humid continental climate region”. This paper presents the laboratory and field studies of constructing the footbridge prototype using UHPFRC composition. In addition, studying the behavior of RC beams using biological self-healing concrete (BSHC). The study is very interesting, and the following comments must be taken for publication possibility:

1.      In the abstract, add (BSHC) after Biological self-healing concrete.

2.      The keywords are very short.

3.      The introduction is very weak. It must be adding more references related to self-healing concrete and its durability performance.

4.      Use the superscript for numbers (kN/m2).

5.      Correct the reference number arranged in the manuscript.

6.      Check the subtitle number.

7.      The authors must present why using the bacterial (B. pseudofirmus and B. cohnii) in the text.

8.      For Fig. 2A, check the dimension of the specimen.

9.      Enhance all figures’ resolution to be readable.

10. Redraw the conclusion of the work. The conclusions must include the important findings of the study and contains the comparisons in ratios.

Author Response

The paper presents “The construction of a footbridge prototype with biological self-healing concrete: A field study in a humid continental climate region”. This paper presents the laboratory and field studies of constructing the footbridge prototype using UHPFRC composition. In addition, studying the behavior of RC beams using biological self-healing concrete (BSHC). The study is very interesting, and the following comments must be taken for publication possibility:

We would like to thank the Referee for reviewing our manuscript and for his/her interest on the topic.

1.In the abstract, add (BSHC) after Biological self-healing concrete.

We revised and corrected the use of acronym BSHC. All changes in the revised version of the manuscript are highlighted in yellow.

2.The keywords are very short.

We added two additional keywords: large-scale demonstration; site trials

3.The introduction is very weak. It must be adding more references related to self-healing concrete and its durability performance.

We thank the Referee for the important comment. As suggested, We have improved the introduction section discussing the environmental impact on bacterial self-healing concrete (BSHC). We also discussed the existing field studies on BSHC as well as limitations of these studies. We also added stronger motivation for the current field study in the Introduction section.

  1. Use the superscript for numbers (kN/m2).

The inaccuracy was corrected.

5.Correct the reference number arranged in the manuscript.

We added 9 new references and corrected the order in the text.

  1. Check the subtitle number.

The inaccuracy was corrected.

  1. The authors must present why using the bacterial (B. pseudofirmus and B. cohnii) in the text.

A sentence in the first paragraph of Section 3.1 was added:

“These bacteria strains were selected due to their high sporulation yields and their ability to survive in the concrete matrix [19].”

8.For Fig. 2A, check the dimension of the specimen.

The inaccuracy was corrected.

9.Enhance all figures’ resolution to be readable.

To minimize the size of the manuscript, the resolution of all figures was reduced. We have uploaded the high-resolution figures separately.

  1. Redraw the conclusion of the work. The conclusions must include the important findings of the study and contains the comparisons in ratios.

As suggested, we have revised the conclusions focusing on the findings from the current study.

Round 2

Reviewer 1 Report

-

Reviewer 2 Report

I wish to thank the author for the revised version so I recommend accepting the paper for publication.